# Bridging Fairness and Uncertainty: Theoretical Insights and Practical Strategies for Equalized Coverage in GNNs

## Abstract

Graph Neural Networks (GNNs) have become indispensable tools in many domains, such as social network analysis, financial fraud detection, and drug discovery. Prior research primarily concentrated on improving prediction accuracy while overlooking how reliable the model predictions are. Conformal prediction on graphs emerges as a promising solution, offering statistically sound uncertainty estimates with a pre-defined coverage level. Despite the promising progress, existing works only focus on achieving model coverage guarantees without considering fairness in the coverage within different demographic groups. To bridge the gap between conformal prediction and fair coverage across different groups, we pose the fundamental question: *Can fair GNNs enable the uncertainty estimates to be fairly applied across demographic groups?* To answer this question, we provide a comprehensive analysis of the uncertainty estimation in fair GNNs employing various strategies. We prove theoretically that fair GNNs can enforce consistent uncertainty bounds across different demographic groups, thereby minimizing bias in uncertainty estimates. Furthermore, we conduct extensive experiments on five commonly used datasets across seven state-of-the-art fair GNN models to validate our theoretical findings. Additionally, based on the theoretical and empirical insights, we identify and analyze the key strategies from various fair GNN models that contribute to ensuring equalized uncertainty estimates. Our work estimates a solid foundation for future exploration of the practical implications and potential adjustments needed to enhance fairness in GNN applications across various domains. For reproducibility, we publish our data and code at https://anonymous.4open.science/r/EqualizedCoverage_CP-9CF8.

## Keywords

Conditional conformal prediction, fairness, graph neural networks

**ACM Reference Format:**

Anonymous Author(s). 2018. Bridging Fairness and Uncertainty: Theoretical Insights and Practical Strategies for Equalized Coverage in GNNs. In *Proceedings of Make sure to enter the correct conference title from your rights confirmation emai (Conference acronym 'XX)*. ACM, New York, NY, USA, 10 pages. https://doi.org/XXXXXXX.XXXXXXX

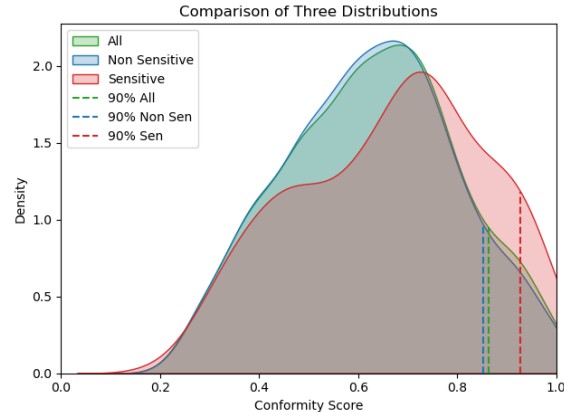

**Figure 1: Comparison of three distributions, areas colored with green denote all samples, blue represents samples from the sensitive group, while red indicates the non-sensitive group. We observe that conformal prediction with the threshold to achieve the desired 90% marginal coverage would cause coverage higher than 90% for the non-sensitive group and lower than 90% for the sensitive group.**

## 1 Introduction

Graph Neural Networks (GNNs) have gained significant importance in recent years and have been extensively used in various domains, such as social network analysis [23], financial fraud detection [5], and drug discovery [24]. As GNNs are increasingly deployed in high-stakes real-world applications, understanding the uncertainty in the predictions made by GNNs becomes vital for enhancing the reliability of GNNs in critical situations [39, 44]. Among several uncertainty quantification techniques on graphs, conformal prediction [4, 10, 36], especially the widely used split conform prediction, emerges as a promising approach due to its efficiency and posthoc nature. It is an easy-to-implement but statistically reliable method to offer uncertainty estimates for any model. Specifically, it aims to construct a confidence interval or prediction set to cover the ground truth with a probability no smaller than a user-specified coverage level, thereby offering critical insights into the reliability of model predictions.

Despite the success of conformal prediction across various domains, most existing work [9, 18, 43] on conformal prediction for graph data primarily focuses on achieving marginal coverage guarantees (i.e., overall coverage for the entire population), neglecting the fact that the coverage could vary across different demographic groups of population defined by a sensitive attribute such as race or gender. Consequently, it may achieve the desired overall coverage by a high coverage above the desired coverage for one demographic group and a low coverage below the desired coverage for another

demographic group, failing to achieve equalized coverage conditioned on the demographic group. Consider a job recommendation system for software engineering positions, where there is a natural imbalance between male and female candidates. Let's say our prediction set claims to be valid with a 5% mis-coverage rate overall. However, upon closer examination, we find that it makes errors with 0% probability for male candidates and 10% probability for female candidates. In this scenario, neither group would be satisfied with the claimed 5% mis-coverage level. This raises a fundamental question: **How can we enable the uncertainty estimates to be fairly applied across demographic groups? How can we achieve equalized coverage?** Figure 1 shows the distribution of the conformity score (the non-agreement between the model prediction and ground truth) of GCN on the CiteSeer [32] dataset. In this figure, the thresholds of conformity scores to achieve 90% coverage for each distribution are different between the sensitive group (red dashed line) and non-sensitive group (blue dashed line), and the threshold of conformity score to achieve 90% marginal coverage for the entire population (green dashed line) lies in between. Consequently, our empirical finding highlights that conformal prediction with the threshold to achieve the desired 90% marginal coverage would cause coverage higher than 90% for the non-sensitive group and lower than 90% for the sensitive group. Thus, it is crucial to investigate how to achieve equalized coverage across different demographic groups.

Achieving equalized coverage in graph learning requires us to re-think where the bias comes from. Existing efforts in fair graph learning mainly study the origin of bias from three aspects [8]: graph structure, input features, and the mechanism of GNN. First, inherent structural biases in graphs (e.g., homophily) [11, 12] make nodes tend to form connections with other similar nodes. For example, a social network may have fewer connections between people from different race groups, leading to biased uncertainty estimates about interactions between people from different race groups. Second, the input features of a GNN model can be biased and unfair [12, 41], reflecting and perpetuating existing bias. For instance, in a job recommendation system, historical data might show a disproportionate number of men in leadership positions, leading to biased feature representations that favor male candidates for executive roles. Lastly, the information aggregation mechanism in GNNs, which combines information from connected nodes, could also amplify bias during the message-passing process, thereby affecting the model fairness [25]. For instance, in a loan approval system modeled as a graph where nodes represent applicants and edges represent similar financial backgrounds, if historically disadvantaged groups are underrepresented or clustered together, the GNN's aggregation process might reinforce existing biases [19]. This could lead to unfair loan approvals, as the model may struggle to accurately assess creditworthiness for underrepresented groups due to limited diverse information propagation. Inspired by prior works trying to achieve equalized predictive performance (e.g., statistical parity [11], equal opportunity [17]), we ask: how can we integrate fairness considerations into the conformal prediction framework for graph data to achieve equalized coverage?

To answer this question, we establish a theoretical connection between equalized coverage and fair GNNs. The key insight is that the coverage discrepancy across different demographic groups is

primarily characterized by a user-specified coverage lower bound $1 - \alpha$ and a group conditional coverage upper bound. The crucial factor influencing this group's conditional coverage upper bound is the probability of a sample being assigned to a particular group. **Consequently, to achieve equalized coverage, it's essential to ensure that the estimated probability of a sample being assigned to each group is equal, which aligns with the objective of fair GNNs.** We provide theoretical proof that fair GNNs are capable of improving equalized coverage across demographic groups. To validate these theoretical findings, we conduct comprehensive experiments on five real-world datasets, evaluating multiple fair GNN models to explore their fairness-aware performance. Furthermore, drawing from both theoretical and empirical insights, we identify and analyze the key strategies employed by various fair GNN models that contribute to achieving equalized coverage.

Our main contributions are summarized as follows:

- **Problem.** We are the first to formulate the problem of equalized coverage in group fair GNNs.
- **Theoretical Insights.** We introduce a theoretical framework for conditional coverage on graphs, offering a mathematical foundation to quantify and understand the coverage guarantees for each demographic group. Our theoretical results show that fair GNNs are helpful in achieving equalized coverage.
- **Extensive Empirical Studies.** We conduct extensive experiments on five datasets with ten GNNs (three without fairness consideration and seven with fairness consideration). We first examine the challenges of achieving equalized uncertainty estimates in GNN models and explore strategies to ensure the uncertainty estimates are consistent and reliable across different demographic groups. Moreover, we conducted a case study to verify the correctness of our theoretical bound on group conditional coverage across a wide range of user-specified coverage.

The rest of this paper is organized as follows. We provide the preliminaries in Section 2, followed by the theoretical insights in Section 3. Section 4 discusses the experimental setup and results, followed by a literature review in Section 5. Finally, we conclude the paper in Section 6.

## 2 Preliminaries

In this section, we introduce preliminaries on Graph Neural Networks (GNNs), conformal prediction, and group fairness, followed by defining the problem of equalized coverage on graphs.

**Notation Convention.** We use upper case calligraphic font letters to denote sets (e.g., $\mathcal{G}$), bold upper case letters to denote matrices (e.g., $\mathbf{A}$), bold lower case letters to denote vectors (e.g., $\mathbf{m}$), and regular lower case letters to denote scalars (e.g., $\alpha$). For matrix indexing, the $i$-th row of a matrix is denoted as its corresponding upper case letter with subscript $i$ (e.g., the $i$-th row of matrix $\mathbf{X}$ is $X_i$), and superscript $^T$ represents matrix transpose.

**Graph Neural Networks (GNNs).** GNNs are designed to efficiently represent graph data by capturing both the structure of the graph and the features of each node. Given an undirected attributed graph $\mathcal{G} = (\mathcal{V}, \mathcal{E})$, where $\mathcal{V} = \{v_1, \cdots, v_N\}$ is the set of nodes and $\mathcal{E}$ is the set of edges. The adjacency matrix is denoted by

$\mathbf{A} \in \mathbb{R}^{N \times N}$, and the node attribute matrix is $\mathbf{X} = [X_1, \cdots, X_N]^T$, where $X_i = X_{v_i}$. Here $N = |\mathcal{V}|$ denotes the number of nodes. GNNs iteratively update node embeddings by aggregating information from their neighbors. At each layer $l$, the embedding of node $v_i$, denoted as $\mathbf{h}_{v_i}^{(l)}$, is updated by using its previous embedding $\mathbf{h}_{v_i}^{(l-1)}$ and the aggregated information from its neighboring nodes $\mathcal{N}(v_i)$. The process is formalized as: $\mathbf{h}_{v_i}^{(l)} = \text{UPD}(\mathbf{h}_{v_i}^{(l-1)}, \text{AGG}(\{\mathbf{h}_{v_j}^{(l-1)} : v_j \in \mathcal{N}(v_i)\}))$, where $\text{AGG}(\cdot)$ is an aggregation function that combines the embeddings of neighboring nodes, and $\text{UPD}(\cdot)$ is an update function that produces the new embedding for node $v_i$.

**Conformal Prediction.** Unlike traditional machine learning models that produce a single prediction, conformal prediction provides valid prediction sets for new samples, accompanied by a guaranteed confidence level. This method is distribution-free and only relies on exchangeability, meaning that it only requires the assumption that the joint distribution remains unchanged when the order of the data set is altered [4, 6]. In this work, we primarily focus on split conformal prediction, which is the most widely-used version of conformal prediction, and is computationally efficient [3, 4]. It divides the training data into two sets: the training set $\mathcal{D}_{\text{train}}$, and the calibration set $\mathcal{D}_{\text{calib}} = \{(X_1, Y_1), \cdots, (X_n, Y_n)\}$. A prediction model $\hat{f}$ (i.e., GNNs) is trained on the training set $\mathcal{D}_{\text{train}}$. Then, given a predefined coverage rate $(1 - \alpha) \in [0, 1]$, it proceeds the following three steps: (1) Conformity score function. The conformity score quantifies the degree of "conformity" of a node with respect to the training data and the overall graph structure. In conformal predictions, we aim to transform a model's heuristic notion of uncertainty into a rigorous one, and the first step is to choose a conformity score function $S(X, Y) \in \mathbb{R}$ that reflects the non-agreement between the prediction of model $\hat{f}$ on $X$ and $Y$. (2) Quantile computation. Compute the $\lceil \frac{(n+1)(1-\alpha)}{n} \rceil$ empirical quantile of the comformity scores $\{S(X_i, Y_i)\}_{i=1}^n$ on the calibration set $\mathcal{D}_{\text{calib}}$ (denoted as $S^*$), where $\lceil \cdot \rceil$ is the ceiling function ($S^*$ is essentially $1 - \alpha$ quantile, but with a small correction). (3) Prediction sets construction. Finally, use the empirical quantile to form the prediction sets for new samples. Specifically, for a test sample, the prediction sets can be formed by the quantile $S^*$ as: $C(X_{\text{test}}) = \{y : S(X_{\text{test}}, y) \leq S^*\}$. If $(X_i, Y_i)_{i=1}^n$ and $(X_{\text{test}}, Y_{\text{test}})$ are exchangeable, then $C(X_{\text{test}})$ contains the ground truth label with predefined coverage rate: $\mathbb{P}(Y_{\text{test}} \in C(X_{\text{test}})) \geq 1 - \alpha$. Specifically, Adaptive Prediction Set (APS) [2] is a widely used conformity score in the task of classification [31] and is utilized in this paper. Its conformity score function computes the cumulative sum of ordered class probabilities (from the most to the least probable class) to the true class. Formally, for a test sample $(X_{\text{test}}, Y_{\text{test}})$, let $\hat{f}_j(X_{\text{test}})$ denotes the predicted probability that $Y_{\text{test}}$ belongs to class $j$, where $j = 1, \cdots, |\mathcal{Y}|$, and $\pi$ be a permutation of the classes so that $\hat{f}_{\pi(1)}(X_{\text{test}}) \geq \hat{f}_{\pi(2)}(X_{\text{test}}) \geq \cdots \geq \hat{f}_{\pi(|\mathcal{Y}|)}(X_{\text{test}})$. The conformity score is defined as the cumulative probability up to the $k$-th most probable class: $S(X_{\text{test}}, k) = \sum_{j=1}^k \hat{f}_{\pi(j)}(X_{\text{test}})$, Then, the prediction set is constructed as $C(X_{\text{test}}) = \{\pi(1), \cdots, \pi(\hat{k})\}$, where $\hat{k}$ is the smallest integer satisfying: $\sum_{j=1}^{\hat{k}} \hat{f}_{\pi(j)}(X_{\text{test}}) \geq S^*$.

**Group Fairness.** Group fairness is a fundamental principle in machine learning that aims to ensure predictive models make unbiased decisions across different demographic groups while maintaining model utility. These groups are defined by protected or sensitive features, such as race, gender, or features that users are usually unwilling to share. The population can be divided into different demographic subgroups based on these features, referred to as sensitive subgroups. Group fairness is then defined upon these sensitive subgroups, generally requiring that the algorithm should not yield discriminary predictions or decisions against individuals from any specific sensitive subgroup. Mathematically, let $A$ be a sensitive attribute with groups $G_1, G_2, \cdots, G_k$, and $\hat{Y}$ be the predicted outcome. One common formulation of group fairness is demographic parity, which requires $P(\hat{Y} = 1|A \in G_1) = P(\hat{Y} = 1|A \in G_2)$ for any two groups $G_1$ and $G_2$. This constraint ensures that the probability of a positive prediction ($\hat{Y} = 1$) is equal across groups. In the context of conformal prediction, satisfying group fairness implies that for any attribute $\forall a \in A$, $\mathbb{P}(Y \in C) \geq 1 - \alpha$ for any user-specified $\alpha \in [0, 1]$, where $C$ is the prediction set given by $X$ and $A$. This formulation extends the concept of group fairness to uncertainty quantification, ensuring that the coverage probability of the prediction set is consistent across all sensitive subgroups.

Estimating uncertainty accurately is essential for enhancing the reliability and trustworthiness of decision-making processes. While recent studies have increasingly concentrated on refining uncertainty estimates, the question of their fairness remains largely neglected. Although some research has touched on fairness, there has been minimal effort to synthesize these concepts comprehensively. Therefore, our research aims to investigate the integration of group fairness models within the framework of conformal prediction, addressing this critical gap in the literature. Here, we introduce the formal problem definition of equalized coverage in group fair GNNs as follows:

PROBLEM 1. ***Equalized Coverage in Group Fair GNNs***
***Input:*** *(i) An undirected attributed graph $\mathcal{G} = (\mathcal{V}, \mathcal{E})$ with the adjacency matrix $\mathbf{A}$, the node attribute matrix $\mathbf{X}$, and the label $\mathbf{Y}$, (ii) an L-layer GNNs model $\hat{f}$, (iii) a sensitive attribute $a$.*
***Output:*** *Equalized coverage for all demographic groups of nodes defined by the sensitive attribute $a$.*

## 3 Fair GNNs Encourage Equalized Coverage

In this section, we introduce the theoretical analysis for fair GNNs in the context of conformal prediction. First, we explore the employment of conformal prediction to graph data, focusing on the exchangeability of nodes in graphs when sensitive attributes are considered. Then we introduce Lemma 2, which offers conditional coverage guarantees for conformal prediction. Finally, we derive the group conditional coverage for each group and demonstrate that fair GNNs can reduce discrepancies of conditional coverage among diverse demographic groups, thus achieving equalized coverage.

ASSUMPTION 1. *For any permutation $\pi_a$ of the calibration and test sets $\mathcal{D}_{calib} \cup \mathcal{D}_{test}$ conditioned on the sensitive attribute $A = a$, the conformity score $S$ of the node $v_j \in \mathcal{D}_{calib} \cup \mathcal{D}_{test}$ remains unchanged. Mathematically,*

$$S(X_j, A_j, Y_j) = S(X_{\pi_a(v_j)}, A_{\pi_a(v_j)}, Y_{\pi_a(v_j)})$$

where node $v_j \in (\mathcal{V}, \mathcal{E})$, $\pi_a(v_j) \in (\mathcal{V}_{\pi_a}, \mathcal{E}_{\pi_a})$, and $(\mathcal{V}, \mathcal{E})$ and $(\mathcal{V}_{\pi_a}, \mathcal{E}_{\pi_a})$ denote the same graph with nodes reordered according to the permutation $\pi_a$, and node $v_j$ and $\pi_a(v_j)$ denote the same node reordered to the permutation $\pi_a$ too. This permutation is applied to the nodes in the calibration and test sets $\mathcal{D}_{calib} \cup \mathcal{D}_{test}$ based on the sensitive attribute $A = a$.

Assumption 1 introduces a crucial requirement for training Graph Neural Networks (GNNs): conditional permutation invariance. This assumption ensures that the conformity score is fair and consistent regardless of the ordering of data within each sensitive subgroup. Essentially, any permutation regarding sensitive subgroups of calibration and test sets doesn't impact the conformity scores for any node in the graph. GNN models typically satisfy this Assumption 1 inherently, as they focus exclusively on the graph's structure and node attributes, disregarding the order of nodes. This is due to the fundamental operation of GNNs, which involves iteratively aggregating information from neighboring nodes and updating node representations, without any dependence on the sequential order of nodes within the graph.

Given Assumption 1, we can derive the coverage bound as presented in the following lemma. Variations of Lemma 2 for conformal prediction have been extensively discussed in prior literature, highlighting its significance in the field. [22, 30, 37].

LEMMA 2. *Suppose the random variables* $Z_1, \cdots, Z_{n+1}$ *are exchangeable on the sensitive attribute* $A_{n+1} = a$, *and* $Q_{1-\alpha}$ *denotes the* $(1-\alpha)(1+1/n)$-*th empirical quantile of* $\{Z_i : 1 \leq i \leq n\}$. *For any* $\alpha \in (0, 1)$,

$$\mathbb{P}\{Z_{n+1} \leq Q_{1-\alpha}|A_{n+1} = a\} \geq 1 - \alpha.$$

*Moreover, if the random variables* $Z_1, \cdots, Z_{n+1}$ *are almost surely distinct, then it holds that*

$$\mathbb{P}\{Z_{n+1} \leq Q_{1-\alpha}|A_{n+1} = a\} \leq 1 - \alpha + \frac{1}{n+1}.$$

*where each* $Z_i = (X_i, A_i, Y_i)$ *is a random variable.*

PROOF SKETCH. Full proof of Lemma 2 are in Appendix A. For the random variables $Z_1, \cdots, Z_n, (X_{n+1}, A_{n+1}, Y_{n+1})$, we rank $S_{n+1}$ among the remaining conformity score $S_1, \cdots, S_n$,

$$\pi(Y_{n+1}) = \frac{1}{n+1} \sum_{i=1}^{n+1} \mathbb{1}\{S_i \leq S_{n+1}\} = \frac{1}{n+1} + \frac{1}{n+1} \sum_{i=1}^{n} \mathbb{1}\{S_i \leq S_{n+1}\}$$

By exchangeability of the data points, when evaluated at $Z_{n+1}$, we see that the constructed statistic $\pi(Y_{n+1})$ is uniformly distributed over the set $\{\frac{1}{n+1}, \frac{2}{n+1}, \cdots, 1\}$, which implies that:

$$\mathbb{P}\left((n+1)\pi(Y_{n+1}) \leq \lceil(1-\alpha)(n+1)\rceil\right) \geq 1 - \alpha,$$

and if the conformity score is almost surely distinct (a weak assumption used to avoid ties when ranking), then

$$\mathbb{P}\left((n+1)\pi(Y_{n+1}) \leq \lceil(1-\alpha)(n+1)\rceil\right) \leq 1 - \alpha + \frac{1}{n+1},$$

By setting $Q_{1-\alpha} = (1-\alpha)(1 + \frac{1}{n})\{Z_i, 1 \leq i \leq n\}$, $\mathbb{P}\{Z_{n+1} \leq Q_{1-\alpha}|A_{n+1} = a\} \geq 1 - \alpha$, and $\mathbb{P}\{Z_{n+1} \leq Q_{1-\alpha}|A_{n+1} = a\} \leq 1 - \alpha + \frac{1}{n+1}$ when $\{Z_i : 1 \leq i \leq n+1\}$ are almost surely distinct. □

The first part of Lemma 2 is a variation of the standard property of all conformal prediction methods, which guarantees the conditional coverage of the prediction sets. For the second part, the surely distinct assumption is quite a weak assumption and is used to avoid ties during the ranking of conformal scores. By employing a random tie-breaking rule, this assumption could be avoided entirely. While Lemma 2 provides the key property of conditional coverage, we further introduce the group conditional coverage for any data distribution in Theorem 3. Group conditional coverage is a more granular measure that quantifies the coverage rate for each individual group, ensuring that the prediction sets maintain the desired coverage level within each group.

THEOREM 3. *If* $(X_i, A_i, Y_i) : 1 \leq i \leq n+1$ *are exchangeable, then the prediction set* $C(X_{n+1}, A_{n+1}) = \{y : S(X_{n+1}, A_{n+1}, y) \leq S^*\}$, *where* $S^*$ *is the* $(1-\alpha)(1+1/n)$-*th empirical quantile of the conformity score evaluated on the calibration set, obeys*

$$1 - \alpha \leq \mathbb{P}\{Y_{n+1} \in C(X_{n+1}, A_{n+1})|A_{n+1} = a\}$$

$$\leq 1 - \alpha + \frac{d}{(n+1)\mathbb{P}(A_{n+1} = a)},$$

*where* $d$ *represents the number of distinct groups based on different values of attribute* $A$. *e.g., the group number (or order) of* $a \in [1, d]$.

PROOF SKETCH. Let $\mathcal{W} = \{\Phi(.)^T\beta : \beta \in \mathbb{R}^d\}$ represents the class of linear functions over the basis $\Phi : \mathcal{X} \rightarrow \mathbb{R}^d$ (the reweighting functions), and let $g$ denotes quantile estimates. Then, for any non-negative $w \in \mathcal{W}$ satisfies $\mathbb{E}_P[w(X, A)] > 0$, then prediction set $C(X_{n+1}, A_{n+1})$ will fulfill the condition $\mathbb{P}_w(Y_{n+1} \in C(X_{n+1}, A_{n+1})) \geq 1 - \alpha$. Additionally, if $(X_1, A_1, Y_1), \cdots, (X_{n+1}, A_{n+1}, Y_{n+1})$ are exchangeable, and $S|(X, A)$ is surely distinct. In that case, we can further assert that for all $w \in \mathcal{W}$, we additionally have the upper bound that (full proof of Theorem 3 are in Appendix A):

$$\mathbb{E}[w(X_{n+1}, A_{n+1})(\mathbb{1}\{Y_{n+1} \in C(X_{n+1}, A_{n+1})\} - (1-\alpha))]$$

$$\leq \frac{d}{n+1}\mathbb{E}\left[\max_{1 \leq i \leq n+1}|w(X_i, A_i)|\right]$$

If $\mathcal{W} = \{\sum_{G \in \mathcal{G}} \beta_G \mathbb{1}\{X \in G\} : \beta_G \in \mathbb{R}\}$, then

$$\mathbb{P}\{Y_{n+1} \in C(X_{n+1}, A_{n+1})|A_{n+1} = a\} \leq 1 - \alpha + \frac{d}{(n+1)\mathbb{P}(X_{n+1} \in G)}$$

$$\leq 1 - \alpha + \frac{d}{(n+1)\mathbb{P}(A_{n+1} = a)}$$
□

According to Theorem 3, we observe that the upper bound of coverage for different groups varies. The crucial factor influencing this variation is the probability of a sample being assigned to a particular group, specifically, $\mathbb{P}(A_{n+1} = a)$. To achieve equalized conditional coverage across groups, it is necessary to ensure that the estimated probability of a sample being assigned to each group is equal. Interestingly, fair group graph neural network (GNN) models inherently aim to eliminate disparities (or discriminations) between groups to ensure group equity. Thus, in this work, we investigate the interplay between group fairness and conformal prediction, exploring how enforcing group fairness in GNNs can potentially contribute to achieving equalized conditional coverage guarantees across groups in the conformal prediction framework.

## 4  Experiments

In this section, we conduct experiments to evaluate our model using 5 datasets with 7 popular fair GNN models and 3 traditional GNN models in terms of uncertainty estimates (coverage and efficiency). We focus on answering the following two questions: **Q1.** Does the state-of-the-art fair GNNs encourage equalized coverage in the task of node classification? And, which designs of fair GNNs are most effective? **Q2.** Does the theoretical bound on group conditional coverage presented in Theorem 3 confirm the empirical results observed in practice?

### 4.1  Experimental Settings

**Datasets.** We conduct experiments on five real-world datasets, including Cora [32], CiteSeer [32], PubMed [32], DBLP [13], Coauthor-Physics [33], and Coauthor-CS [33]. The detailed data statistics are provided in Table 1.

**Table 1: Statistics of real-world graph benchmarks. The number (n) next to the sensitive attribute label indicates how many values the sensitive attribute may take on.**

| Types | Datasets | # Nodes | # Edges | # Features | Sensitive | # Label |
|-------|----------|---------|---------|------------|-----------|---------|
| Citation | Cora | 2,708 | 10,556 | 1,433 | Topic (2) | 7 |
|  | CiteSeer | 3,327 | 9,104 | 3,703 | Topic (2) | 6 |
|  | PubMed | 19,717 | 88,648 | 500 | Topic (2) | 3 |
| Coauthor | CS | 18,333 | 163,788 | 6,805 | Topic (2) | 15 |
|  | Physics | 34,493 | 495,924 | 8,415 | Topic (2) | 5 |

**Group Fairness GNNs.** We compare classical GNNs with popular group fairness GNNs. Among the classical models, GCN [21] employs a spectral-based approach to aggregate and update node features through graph neighborhoods. GraphSAGE [16] introduces an inductive framework that learns aggregation functions to generate node embeddings for unseen data. And GAT [34] utilizes attention mechanisms to assign different importance to neighboring nodes during feature aggregation. In the realm of fair GNNs, we examine several approaches: Fairwalk [28] is a modified random walk method that aims to generate fair node embeddings by adjusting transition probabilities based on sensitive attributes. CrossWalk [20] enhances graph embeddings by encouraging connections between nodes from different demographic groups during the random walk. GEAR [27] learns fair graph embeddings by optimizing a fairness-aware objective function alongside the traditional embedding loss. UGE [40] mitigates bias by reweighting edge sampling probabilities based on node degrees and sensitive attributes. NIFTY [1] improves fairness in graph neural networks by introducing a fairness-aware loss term during training. FairVGNN [41] incorporates fairness constraints into the learning process to generate fair node representations. And BeMap [25] adjusts the aggregation process to reduce unfairness in node representations. These methods each tackle fairness in graph learning from different angles, aiming to address biases in various aspects, from graph structural bias elimination (e.g., Fairwalk, CrossWalk), input feature correction (e.g., NIFTY, FairVGNN), to modifying the mechanism of GNNs (e.g., BeMap), we aim to explore their strengths and weaknesses in different settings.

**Parameter Settings.** Unless stated otherwise, we follow the default hyperparameter settings in the released code of the corresponding publications. This ensures that our experiments remain consistent and comparable. However, the parameters of some methods are optimized for the task of binary classification, which may lead to less consistent performance when deployed in different scenarios.

**Evaluation Metrics.** We consider the task of node classification. The goal is to ensure valid coverage while minimizing the inefficiency as much as possible, Furthermore, to promote group fairness, we also aim to reduce disparities in coverage and inefficiency between different demographic groups. We evaluate performance using four key metrics: coverage, inefficiency, coverage difference, and inefficiency difference. Specifically, the coverage is defined as the proportion of instances for which the true label is included in the prediction set, given by Coverage $:= \frac{1}{n} \sum_{i=1}^{n} \mathbb{1}(Y_i \in C(X_i, A_i))$. Inefficiency refers to the average size of the prediction set: Ineff $:= \frac{1}{n} \sum_{i=1}^{n} |C(X_i, A_i)|$. The coverage difference measures the disparity between groups, defined as $\Delta$Coverage $:= |\text{Coverage}(a = 1) - \text{Coverage}(a = 0)|$, while the inefficiency difference quantifies the disparity in prediction set size between groups, defined as $\Delta$Ineff $:= |\text{Ineff}(a = 1) - \text{Ineff}(a = 0)|$. It is worth noting that inefficiency is as critical as coverage—if inefficiency is disregarded, methods could achieve 100% coverage by including all candidate labels, which would result in maximum inefficiency. Thus, balancing both metrics is crucial for achieving meaningful and fair predictions.

**Implementation Details.** We implement traditional GNNs with PyG. All the datasets we used can be found in PyG. For those datasets that do not provide splits, we split them follow a standard evaluation procedure, where we randomly split dataset into 20%, 10%, and 70% for model training, validation, and calibration and testing. We use the same splits for all the models for fair comparison. The experiments are performed on a machine with Nvidia GPU A100 (80G), the Python version is 3.9.19, the Pytorch version is 1.12.0, the CUDA version is 12.4, and the PyG version is 2.1.0. We publish our data and code for reproducibility. [1]

### 4.2  (Q1) Quantitative Evaluation on Fair GNNs

In this subsection, we conduct a quantitative analysis of the results of various methods. The main results on the utility (coverage and inefficiency) and fairness ($\Delta$Coverage and $\Delta$Inefficiency) are presented in Table 2. We can notice that: (1). For classical GNN models, Graphsage demonstrates superior performance in terms of fairness compared to its counterparts on large-scale datasets. However, it falls short in utility performance, characterized by significant inefficiencies. Similar trends are observed in fair GNN models like NIFTY and FairVGNN, which pay more attention to fairness. (2). FairWalk, CrossWalk, and UGE primarily focus on reducing the structural bias. Both FairWalk and CrossWalk are based on the Node2Vec model and enhance it by introducing constraints to the random walk process, which helps generate less biased predictions. UGE goes a step further by not only addressing structural bias but also incorporating a regularization technique to minimize discrepancies in predictions with and without the sensitive attribute. These approaches have demonstrated more consistent performance in

---

[1] https://anonymous.4open.science/r/EqualizedCoverage_CP-9CF8

**Table 2: Results of conformal prediction with GNNs models. The result takes the average and standard deviation across 5 GNN runs with the pre-defined coverage (e.g., $1 - \alpha$=90%).**

| Dataset | Method | Coverage (%) | Inefficiency ↓ | Group Coverage(%) | | ΔCoverage(%) ↓ | ΔInefficiency ↓ |
|---|---|---|---|---|---|---|---|
| | | | | Coverage (Sens.) | Coverage (Non-sens.) | | |
| Cora | GCN | 91.12 ± 1.87 | 3.59 ± 0.35 | 96.00 ± 1.89 | 90.57 ± 2.06 | 5.43 ± 2.08 | 0.45 ± 0.07 |
| | GraphSage | 90.84 ± 1.45 | 4.68 ± 0.39 | 94.32 ± 3.78 | 90.46 ± 1.34 | 3.86 ± 3.88 | 0.14 ± 0.10 |
| | GAT | 89.98 ± 1.05 | 3.34 ± 0.57 | 93.26 ± 2.73 | 89.61 ± 0.89 | 3.65 ± 2.92 | 0.38 ± 0.13 |
| | FairWalk | 90.38 ± 1.80 | 4.08 ± 0.79 | 91.65 ± 6.39 | 90.24 ± 1.96 | 1.41 ± 3.13 | 0.15 ± 0.11 |
| | CrossWalk | 90.55 ± 1.64 | 3.89 ± 0.45 | 92.37 ± 3.95 | 90.35 ± 2.02 | 2.02 ± 3.23 | 0.12 ± 0.10 |
| | UGE | 89.77 ± 0.21 | 3.92 ± 0.11 | 90.88 ± 1.40 | 89.64 ± 0.07 | 1.29 ± 1.20 | 0.14 ± 0.12 |
| | GEAR | 90.19 ± 1.05 | 3.45 ± 0.19 | 89.89 ± 2.74 | 90.22 ± 0.87 | 0.33 ± 1.28 | 0.14 ± 0.02 |
| | NIFTY | 89.98 ± 0.01 | 5.22 ± 0.59 | 87.82 ± 1.65 | 90.24 ± 0.01 | 2.42 ± 1.97 | 0.03 ± 0.03 |
| | FairVGNN | 89.66 ± 0.11 | 4.85 ± 1.02 | 87.97 ± 0.38 | 89.85 ± 0.13 | 1.88 ± 2.91 | 0.05 ± 0.04 |
| | BeMap | 91.60 ± 0.67 | 4.15 ± 0.01 | 94.12 ± 2.45 | 91.33 ± 0.42 | 2.79 ± 2.00 | 0.34 ± 0.08 |
| CiteSeer | GCN | 90.84 ± 1.22 | 2.86 ± 0.11 | 87.32 ± 1.61 | 91.16 ± 1.35 | 3.84 ± 2.48 | 0.29 ± 0.11 |
| | GraphSage | 93.01 ± 6.62 | 4.60 ± 1.23 | 96.43 ± 3.57 | 92.69 ± 6.90 | 3.74 ± 3.33 | 0.10 ± 0.08 |
| | GAT | 91.81 ± 2.42 | 2.76 ± 0.39 | 90.54 ± 2.32 | 91.93 ± 2.53 | 1.39 ± 2.10 | 0.21 ± 0.13 |
| | FairWalk | 93.23 ± 5.71 | 5.13 ± 0.65 | 93.93 ± 6.42 | 93.17 ± 0.16 | 0.76 ± 3.14 | 0.07 ± 0.08 |
| | CrossWalk | 90.51 ± 1.95 | 4.67 ± 0.27 | 89.11 ± 4.63 | 90.64 ± 2.15 | 1.53 ± 4.17 | 0.07 ± 0.07 |
| | UGE | 90.88 ± 0.08 | 4.39 ± 0.14 | 91.61 ± 0.41 | 90.81 ± 0.99 | 0.79 ± 3.82 | 0.09 ± 0.08 |
| | GEAR | 88.24 ± 0.35 | 4.03 ± 0.40 | 87.20 ± 2.08 | 88.33 ± 0.58 | 1.13 ± 0.72 | 0.02 ± 0.02 |
| | NIFTY | 88.03 ± 1.12 | 4.89 ± 0.33 | 86.86 ± 5.61 | 88.13 ± 0.70 | 1.27 ± 3.78 | 0.04 ± 0.10 |
| | FairVGNN | 89.01 ± 0.68 | 4.11 ± 0.55 | 88.84 ± 3.13 | 89.03 ± 0.62 | 0.19 ± 1.21 | 0.06 ± 0.08 |
| | BeMap | 90.33 ± 1.63 | 3.15 ± 0.28 | 91.43 ± 2.14 | 90.24 ± 1.58 | 1.19 ± 0.57 | 0.19 ± 0.07 |
| PubMed | GCN | 88.48 ± 0.38 | 1.55 ± 0.04 | 91.31 ± 0.53 | 88.22 ± 0.37 | 3.09 ± 0.45 | 0.21 ± 0.03 |
| | GraphSage | 91.33 ± 8.45 | 1.95 ± 0.97 | 91.93 ± 7.79 | 91.28 ± 8.51 | 0.65 ± 0.76 | 0.08 ± 0.07 |
| | GAT | 93.34 ± 0.55 | 1.48 ± 0.04 | 94.47 ± 0.45 | 93.23 ± 0.59 | 1.24 ± 0.14 | 0.24 ± 0.03 |
| | FairWalk | 88.07 ± 0.58 | 1.77 ± 0.02 | 85.58 ± 2.09 | 88.30 ± 0.47 | 2.71 ± 1.64 | 0.05 ± 0.02 |
| | CrossWalk | 87.61 ± 1.19 | 1.75 ± 0.04 | 84.87 ± 1.01 | 87.87 ± 1.24 | 3.00 ± 0.74 | 0.05 ± 0.01 |
| | UGE | 90.62 ± 0.13 | 1.93 ± 0.05 | 88.01 ± 1.02 | 90.87 ± 0.23 | 2.86 ± 1.16 | 0.06 ± 0.04 |
| | GEAR | 89.41 ± 0.87 | 1.42 ± 0.05 | 91.93 ± 1.25 | 89.02 ± 0.83 | 2.91 ± 0.42 | 0.19 ± 0.07 |
| | NIFTY | 90.65 ± 0.15 | 2.98 ± 0.01 | 99.41 ± 0.32 | 89.67 ± 0.13 | 0.26 ± 0.30 | 0.01 ± 0.01 |
| | FairVGNN | 90.18 ± 0.60 | 1.80 ± 0.18 | 88.61 ± 0.06 | 90.33 ± 0.59 | 1.72 ± 1.67 | 0.19 ± 0.04 |
| | BeMap | 89.44 ± 1.31 | 1.71 ± 0.05 | 91.05 ± 0.42 | 89.29 ± 1.39 | 1.77 ± 0.97 | 0.11 ± 0.07 |
| Physics | GCN | 89.68 ± 0.05 | 3.02 ± 0.15 | 73.79 ± 3.38 | 91.13 ± 0.74 | 17.33 ± 3.83 | 0.84 ± 0.10 |
| | GraphSage | 99.96 ± 0.01 | 4.91 ± 0.01 | 99.96 ± 0.01 | 99.96 ± 0.01 | 0.01 ± 0.01 | 0.04 ± 0.01 |
| | GAT | 89.73 ± 0.78 | 2.06 ± 0.18 | 90.12 ± 7.81 | 89.70 ± 1.21 | 0.42 ± 0.66 | 0.67 ± 0.28 |
| | FairWalk | 90.20 ± 0.26 | 2.92 ± 0.05 | 77.89 ± 0.17 | 91.32 ± 0.15 | 13.43 ± 1.96 | 0.72 ± 0.07 |
| | CrossWalk | 89.75 ± 0.47 | 2.84 ± 0.04 | 75.52 ± 0.18 | 91.05 ± 0.36 | 15.53 ± 1.53 | 0.69 ± 0.06 |
| | UGE | 89.44 ± 0.01 | 1.75 ± 0.02 | 80.38 ± 0.01 | 90.28 ± 0.01 | 9.89 ± 0.01 | 0.51 ± 0.01 |
| | GEAR | 90.86 ± 0.32 | 2.23 ± 0.02 | 85.38 ± 1.76 | 91.36 ± 0.19 | 5.98 ± 1.57 | 0.79 ± 0.02 |
| | NIFTY | 90.31 ± 1.40 | 3.64 ± 0.62 | 98.75 ± 0.35 | 89.54 ± 1.50 | 9.21 ± 1.15 | 0.04 ± 0.06 |
| | FairVGNN | 93.32 ± 0.05 | 4.29 ± 0.62 | 99.01 ± 0.94 | 92.80 ± 5.55 | 6.21 ± 4.61 | 0.07 ± 0.19 |
| | BeMap | 89.67 ± 0.25 | 2.71 ± 0.14 | 83.37 ± 5.11 | 90.24 ± 0.74 | 6.87 ± 5.86 | 0.30 ± 0.32 |
| CS | GCN | 90.10 ± 0.93 | 7.99 ± 0.23 | 93.36 ± 0.36 | 89.76 ± 1.01 | 3.59 ± 0.89 | 0.67 ± 0.43 |
| | GraphSage | 99.97 ± 0.01 | 14.66 ± 0.04 | 99.90 ± 0.01 | 99.97 ± 0.01 | 0.07 ± 0.01 | 0.12 ± 0.07 |
| | GAT | 89.61 ± 0.01 | 6.47 ± 0.02 | 98.55 ± 0.04 | 88.76 ± 0.05 | 9.79 ± 0.40 | 1.65 ± 0.77 |
| | FairWalk | 91.52 ± 0.76 | 7.08 ± 0.32 | 91.13 ± 1.37 | 91.55 ± 0.69 | 0.42 ± 0.56 | 0.55 ± 0.21 |
| | CrossWalk | 90.66 ± 1.01 | 7.50 ± 0.39 | 89.23 ± 1.80 | 91.04 ± 1.04 | 1.81 ± 0.98 | 0.49 ± 0.08 |
| | UGE | 91.70 ± 0.88 | 4.07 ± 0.28 | 94.41 ± 0.29 | 91.44 ± 0.94 | 2.97 ± 0.65 | 0.52 ± 0.06 |
| | GEAR | 89.14 ± 0.40 | 6.30 ± 0.26 | 91.71 ± 0.87 | 88.90 ± 0.35 | 2.81 ± 0.58 | 0.13 ± 0.06 |
| | NIFTY | 89.00 ± 0.76 | 9.48 ± 2.13 | 91.98 ± 1.66 | 88.71 ± 0.67 | 3.27 ± 1.95 | 0.18 ± 0.14 |
| | FairVGNN | 89.01 ± 0.65 | 10.54 ± 1.30 | 93.92 ± 2.22 | 88.53 ± 0.53 | 5.39 ± 2.23 | 0.46 ± 0.14 |
| | BeMap | 90.37 ± 0.58 | 5.37 ± 0.10 | 92.47 ± 0.30 | 90.17 ± 0.95 | 2.30 ± 0.42 | 0.56 ± 0.12 |

terms of ΔCoverage and ΔIneff across various datasets, indicating superior effectiveness in comprehensive evaluations. (3). GEAR,

NIFTY, and FairVGNN all interact with input features to promote fairness. NIFTY focuses on maximizing the similarities between

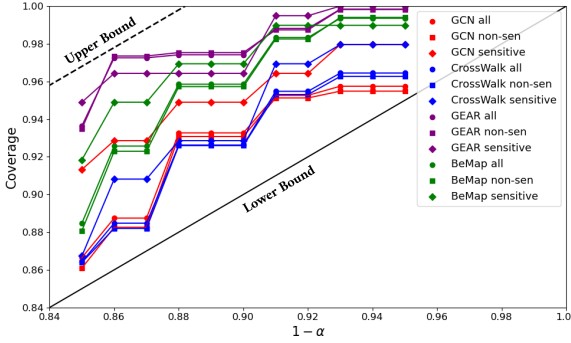

**Figure 2: Qualitative evaluation on group-conditional coverage bound w.r.t. 1- $\alpha$.**

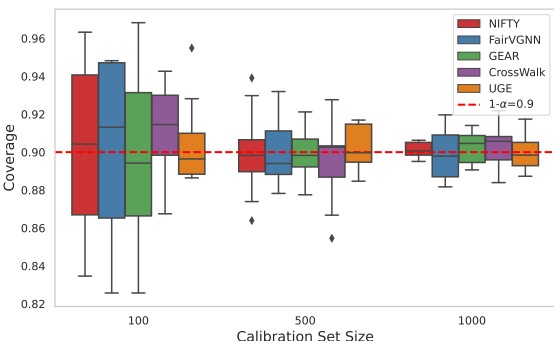

**Figure 3: Boxplot of the marginal coverage w.r.t. the size of calibration set.**

outputs from the original graph and the augmented graph, which involves minor random perturbations to node attributes and/or edges and modifications of the sensitive attribute while also introducing layerwise weight normalization to enhance message passing. GEAR aims to minimize the discrepancy between predictions from the original graph and counterfactual data augmentation, which alters sensitive attributes of nodes and their neighbors. FairVGNN constructs a fair view of features by automatically identifying and masking sensitive-correlated features and adjusting the encoder's weight to avoid using such sensitive-related features. However, while these methods prioritize fairness, they often fall short of generating efficient predictions, particularly in large-scale datasets. (4) BeMap investigates the problem of bias amplification in message passing and leverages a balance-aware sampling strategy to balance the number of the 1-hop neighbors of each node among different demographic groups. While it may not consistently deliver top performance in all settings, its performance remains stable and reliable. Finally, these findings highlight the need to prioritize addressing structural bias in the process of reducing bias in conformal predictions. Attention should then shift to examining the mechanisms of message passing and the corresponding input features.

## 4.3 (Q2) Qualitative Evaluation on Group-conditional Coverage Bound

We also conduct a qualitative analysis of the results. Specifically, we first plot the coverage of some representative methods to empirically prove the validity of Theorem 3. Then we explore the differences in conformity scores across different methods to gain deep insights into their performance. Figure 2 presents a series of plots on the dataset of Cora, each showing how coverage changes with the change of alpha value for a particular method within each group. Each method is shown in the same color, and different shapes are used to show groups within those methods. This figure helps to identify trends and variations in coverage as the alpha value is adjusted across different methods and groups. The solid line illustrates the lower bound, while the dashed line represents the upper bound. We can notice that these lines are within the lower bound and upper bound, which further verifies Theorem 3. Additionally, it is worth noting that while the theoretical guarantee of coverage holds, the size of the calibration set can

lead to variations in observed coverage. Vladimir Vovk [35] first introduces that the distribution of coverage follows a Beta distribution: $\mathbb{P}\{Y_{n+1} \in C(X_{n+1})|\{(X_i, Y_i)\}_{i=1}^{n}\} \sim \text{Beta}(n+1-l, l)$, where $l = \lfloor (n+1)\alpha \rfloor$. Although the bound may fluctuate slightly, if the size of the calibration set $n$ (e.g., 1000) is carefully chosen, the coverage will concentrate sufficiently tightly around $1-\alpha$. This phenomenon can be observed in Figure 3. We also observe that the sensitivity to calibration set size varies among methods: some exhibit significant coverage fluctuations when the calibration set deviates from its optimal size (e.g., NFITY, FairVGNN, etc.), while others demonstrate smoother performance with smoother coverage across different calibration set sizes (e.g., CrossWalk, UGE).

We also plot the distribution of conformity scores for different groups in Figure 4, green represents all samples, red denotes the samples from the sensitive group, and blue indicates the samples from the non-sensitive group. A vertical dashed line is utilized to denote the 90% quantile of each group's distribution with the corresponding color. Due to space limits, we choose some representative methods presented in this figure. We observe that: (1) The 90% quantile difference among different groups using fairness methods is minor when compared to GCN. This indicates that these methods exhibit smaller differences in coverage and inefficiency relative to GCN, suggesting a more consistent performance across the groups. (2) However, different fair GNN methods perform differently in the distribution of conformity scores, and we can notice that the differences in group distribution between CrossWalk, GEAR, and BeMap are smaller than those between NIFTY and FairVGNN, which can partly explain why they have better performance. (3) The shape of distribution also matters. When the distribution is more skewed and closer to the right side, the inefficiency of this method is lower (e.g., CrossWalk, BeMap), and when this distribution is more uniform, or smoother, its inefficiency is higher (e.g., NIFTY, FairVGNN).

## 5 Related Work

**Fairness in Graph Learning.** As GNNs have become increasingly important in recent years and have been successfully used in many areas, the fairness issues in GNNs have been actively studied. Bose and Hamilton [7] proposed an adversarial approach to learn fair graph embeddings by adding an adversary network to the embedding to remove sensitive attribute information. Building upon the

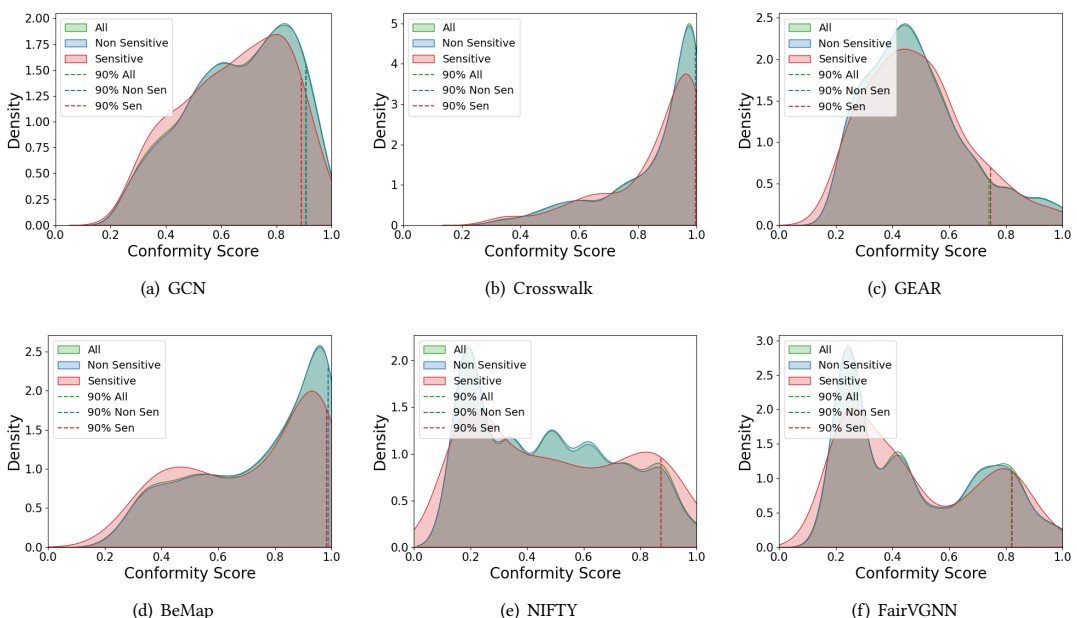

(a) GCN            (b) Crosswalk            (c) GEAR

(d) BeMap            (e) NIFTY            (f) FairVGNN

**Figure 4: Distribution of conformity scores.**

Node2Vec [15] model, FairWalk [28] and CrossWalk [20] introduce constraints into the random walk process, enhancing fairness in representation learning. UGE [40] extends this pursuit by not only addressing structural biases but also integrating regularization techniques to minimize prediction disparities concerning sensitive attributes. In terms of counterfactual fairness, NIFTY [1] focuses on maximizing similarities between outputs from original and augmented graphs through minor perturbations in node attributes and/or edges. Similarly, GEAR [27] aims to minimize prediction discrepancies between original and counterfactual data-augmented graphs by altering sensitive attributes of nodes and their neighbors. Moreover, BeMap [25] explores bias amplification in message passing, employing a balance-aware sampling approach to ensure equitable representation of 1-hop neighbors across demographic groups.

**Conformal prediction.** Conditional coverage using conformal prediction has primarily been investigated in the context of i.i.d. (independent and identically distributed) data. Gibbs et al. [14] studied the problem of constructing distribution-free prediction sets with finite-sample conditional guarantees. Romano et al. [29] introduced a method to construct unbiased prediction intervals for regression tasks, while Wang et al. [38] further refined this approach to guarantee equal coverage rates across groups with finer granularity. Despite these advancements, the application of conformal prediction to graph-structured data remains largely unexplored. Wijegunawardana et al. [42] first apply conformal prediction on graphs. Zargarbashi et al. [43] study the exchangeability under the transductive setting and propose a diffusion-based method for improving efficiency. Lunde et al. [26] studies exchangeability in network regression for non-conformity scores based on various network structures. Huang et al. [18] first study the transductive

setting where certain exchangeability property holds and propose a regularizer to reduce the inefficiency. However, little attention has been paid to achieving equalized coverage within different demographic groups on graphs. Our work aims to bridge this gap by extending conformal prediction techniques to graph-structured data while focusing on fairness considerations across diverse demographic groups.

## 6 Conclusion

In this paper, we introduce the concept of equalized coverage in fair GNNs through a novel theoretical framework for conditional coverage. This framework provides a mathematical foundation for quantifying coverage guarantees across different groups, paving the way for equitable conformal prediction methods in fair GNNs. We explore the application of conformal prediction to graph data, focusing on node exchangeability with respect to sensitive attributes, and derive a theoretical bound showing how fair GNNs can reduce discrepancies in conditional coverage, thereby promoting equalized coverage across diverse groups. Our contributions include a rigorous theoretical analysis bridging the gap between fair GNNs and equalized coverage and comprehensive empirical studies validating the theoretical bound by evaluating the uncertainty estimates of seven fair GNN models and three traditional GNN models. Additionally, we identify strategies that promote equalized coverage, offering insights for future research and practical applications. By addressing the critical issue of fairness in GNNs and conformal prediction, our work promotes trustworthiness in machine learning applications across domains and lays a solid foundation for further exploration of fairness in GNNs. For reproducibility, the data and code are available at https://anonymous.4open.science/r/EqualizedCoverage_CP-9CF8.

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

## A Proofs of theoretical results

PROOF OF LEMMA 2. For the random variables $Z_1, \cdots, Z_n$, $(X_{n+1}, A_{n+1}, Y_{n+1})$, we rank $S_{n+1}$ (simplified version of $S(X_{n+1}, A_{n+1}, Y_{n+1})$) among the remaining conformity score $S_1, \cdots, S_n$, computing

$$\pi(Y_{n+1}) = \frac{1}{n+1} \sum_{i=1}^{n+1} \mathbb{1}\{S_i \leq S_{n+1}\} = \frac{1}{n+1} + \frac{1}{n+1} \sum_{i=1}^{n} \mathbb{1}\{S_i \leq S_{n+1}\}$$

By exchangeability of the data points, when evaluated at $Z_{n+1}$, the rank of $Y_{n+1}$ is uniformly distributed over the values $Y_1, \cdots, Y_{n+1}$, which means that:

$$\mathbb{P}\left(Y_{n+1} \text{ among the } \lceil(1-\alpha)(n+1)\rceil \text{ smallest of } Y_1, \cdots, Y_{n+1}\right)$$
$$\geq 1 - \alpha$$

which equivalent to that the constructed statistic $\pi(Y_{n+1})$ is uniformly distributed over the set $\{\frac{1}{n+1}, \frac{2}{n+1}, \cdots, 1\}$, and:

$$\mathbb{P}\left((n+1)\pi(Y_{n+1}) \leq \lceil(1-\alpha)(n+1)\rceil\right) \geq 1 - \alpha,$$

and if the conformity score is almost surely distinct (a weak assumption used to avoid ties when ranking), then

$$\mathbb{P}\left((n+1)\pi(Y_{n+1}) \leq \lceil(1-\alpha)(n+1)\rceil\right) \leq 1 - \alpha + \frac{1}{n+1},$$

Considering that $Y_{n+1}$ can never be strictly larger than itself,

$$\mathbb{P}\left(Y_{n+1} \text{ is among the } \lceil(1-\alpha)(n+1)\rceil \text{ smallest of } Y_1, \cdots, Y_{n+1}\right)$$
$$\geq 1 - \alpha,$$

is equivalent to:

$$\mathbb{P}\left(Y_{n+1} \text{ is among the } \lceil(1-\alpha)(n+1)\rceil \text{ smallest of } Y_1, \cdots, Y_n\right)$$
$$\geq 1 - \alpha,$$

Thus, the above equation can be transformed to,

$$\mathbb{P}\left(n \cdot \pi(Y_{n+1}) \leq \lceil(1-\alpha)(n+1)\rceil\right) \leq 1 - \alpha + \frac{1}{n+1},$$

By setting $Q_{1-\alpha} = (1-\alpha)(1 + \frac{1}{n})\{Z_i, 1 \leq i \leq n\}$,

$$\mathbb{P}\{Z_{n+1} \leq Q_{1-\alpha} | A_{n+1} = a\} \geq 1 - \alpha,$$

and

$$\mathbb{P}\{Z_{n+1} \leq Q_{1-\alpha} | A_{n+1} = a\} \leq 1 - \alpha + \frac{1}{n+1}$$

when $\{Z_i : 1 \leq i \leq n+1\}$ are almost surely distinct. $\square$

PROOF OF THEOREM 3. Let $\mathcal{W} = \{\Phi(.)^T \beta : \beta \in \mathbb{R}^d\}$ represents the class of linear functions over the basis $\Phi : \mathcal{X} \to \mathbb{R}^d$ (the reweighting functions), and let $g$ denotes quantile estimates. Then, for any non-negative $w \in \mathcal{W}$ sttisfies $\mathbb{E}_P[w(X, A)] > 0$, then prediction set $C(X_{n+1}, A_{n+1})$ will fulfill the condition $\mathbb{P}_w(Y_{n+1} \in C(X_{n+1}, A_{n+1})) \geq 1 - \alpha$. Additionally, if $(X_1, A_1, Y_1), \cdots,$ $(X_{n+1}, A_{n+1}, Y_{n+1})$ are exchangeable, and $S|(X, A)$ is surely distinct, we can further assert that for all $w \in \mathcal{W}$, we additionally have the upper bound that $\mathbb{E}[w(X_{n+1}, A_{n+1})(\mathbb{1}\{Y_{n+1} \in C(X_{n+1}, A_{n+1})\} - (1-\alpha))] \leq \frac{d}{n+1} \mathbb{E}\left[\max_{1 \leq i \leq n+1} |w(X_i, A_i)|\right]$.

$$\mathbb{E}\left[w(X_{n+1}, A_{n+1})(\mathbb{1}\{Y_{n+1} \in C(X_{n+1}, A_{n+1})\} - (1-\alpha))\right]$$
$$= \mathbb{E}\left[w(X_{n+1}, A_{n+1})(\alpha - \mathbb{1}\{Y_{n+1} \notin C(X_{n+1}, A_{n+1})\})\right]$$
$$= \mathbb{E}\left[w(X_{n+1}, A_{n+1})(\alpha - \mathbb{1}\{S_{n+1} > g_{S_{n+1}}(X_{n+1}, A_{n+1})\})\right]$$

Additionally, given that $g_{S_{n+1}}$ is symmetrically fitted and therefore invariant to permutations of the input data, it follows that the set $\{X_i, A_i, g_{S_{n+1}}(X_i, A_i), S_i\}$ is exchangeable. Consequently, it follows that:

$$\mathbb{E}\left[w(X_{n+1}, A_{n+1})(\alpha - \mathbb{1}\{S_{n+1} > g_{S_{n+1}}(X_{n+1}, A_{n+1})\})\right]$$
$$= \mathbb{E}\left[\frac{1}{n+1} \sum_{i=1}^{n+1} w(X_{n+1}, A_{n+1})(\alpha - \mathbb{1}\{S_i > g_{S_{n+1}}(X_i, A_i)\})\right]$$
$$\leq \mathbb{E}\left[\frac{1}{n+1} \sum_{i=1}^{n+1} \left(w(X_{n+1}, A_{n+1})(\alpha - s_i^*)\mathbb{1}\{S_i = g_{S_{n+1}}(X_i, A_i)\}\right)\right]$$

Finally, given that $\alpha - s_i^* \in [0, 1]$, we can establish a bound for the expectation as follows:

$$\mathbb{E}[w(X_{n+1}, A_{n+1})(\mathbb{1}\{Y_{n+1} \in C(X_{n+1}, A_{n+1})\} - (1-\alpha))]$$
$$\leq \mathbb{E}\left[\frac{1}{n+1} \sum_{i=1}^{n+1} \left(w(X_{n+1}, A_{n+1})\mathbb{1}\{S_i = g_{S_{n+1}}(X_i, A_i)\}\right)\right]$$

Furthermore, considering that $g_{S_{n+1}}(X_i, A_i)$ is defined as $\Phi(X_i, A_i)^T \hat{\beta}$, where $\hat{\beta} \in \mathbb{R}^d$, we could deduce that:

$$\mathbb{P}\left(\frac{1}{n+1} \sum_{i=1}^{n+1} \left(\mathbb{1}\{S_i = g_{S_{n+1}}(X_i, A_i)\} > d | (X_1 A_1), \cdots, (X_{n+1}, A_{n+1})\right)\right)$$
$$= \mathbb{P}\left(\exists 1 \leq j_1 < \cdots < j_{d+1} \leq n+1\right)$$

such that $\forall i, S_{j_i} = g_{S_{n+1}}(X_{j_i}, A_{j_i}) | (X_k, A_k)_{k=1}^{n+1}$
$$\leq \sum_{j_1 < \cdots < j_{d+1}} \mathbb{P}\left(\exists \beta \in \mathbb{R}^d\right)$$

such that $\forall i, S_{j_i} = \Phi(X_{j_i}, A_{j_i})^T \beta | (X_k, A_k)_{k=1}^{n+1}$
$$\leq \sum_{j_1 < \cdots < j_{d+1}} \mathbb{P}\left((S_{j_1}, \cdots, S_{j_{d+1}}) \in \text{RowSpace}([\Phi(\cdot)]) | (X_k, A_k)_{k=1}^{n+1}\right)$$

where $\text{RowSpace}([\Phi(\cdot)])$ is the abbreviation of $\text{RowSpace}([\Phi(X_{j_i}, A_{j_i}), \cdots, \Phi(X_{j_{d+1}}, A_{j_{d+1}})]^T)$ is a $d$-dimensional subspace of $\mathbb{R}^{d+1}$.

Considering that $(S_{j_1}, \cdots, S_{j_{d+1}}) | ((X_k, A_k)_{k=1}^{n+1})$ are independent and surely distinct, so,

$$\mathbb{P}\left(\frac{1}{n+1} \sum_{i=1}^{n+1} \left(\mathbb{1}\{S_i = g_{S_{n+1}}(X_i, A_i)\} > d | (X_1, A_1) \cdots, (X_{n+1}, A_{n+1})\right)\right)$$
$$= 0$$

Then, we could conclude that with probability 1,

$$\frac{1}{n+1} \sum_{i=1}^{n+1} \left(\mathbb{1}\{S_i = g_{S_{n+1}}(X_i, A_i)\}\right) \leq \frac{d}{n+1}.$$

Then,

$$\mathbb{E}[w(X_{n+1}, A_{n+1})(\mathbb{1}\{Y_{n+1} \in C(X_{n+1}, A_{n+1})\} - (1-\alpha))]$$
$$\leq \frac{d}{n+1} \mathbb{E}\left[\max_{1 \leq i \leq n+1} |w(X_i, A_i)|\right]$$

If $\mathcal{W} = \{\sum_{G \in \mathcal{G}} \beta_G \mathbb{1}\{X \in G\} : \beta_G \in \mathbb{R}\}$, then

$$\mathbb{P}\{Y_{n+1} \in C(X_{n+1}, A_{n+1}) | A_{n+1} = a\}$$
$$\leq 1 - \alpha + \frac{d}{(n+1)\mathbb{P}(X_{n+1} \in G)}$$
$$\leq 1 - \alpha + \frac{d}{(n+1)\mathbb{P}(A_{n+1} = a)}$$

$\square$

