# OpenReview forum: "Bridging Fairness and Uncertainty: Theoretical Insights and Practical Strategies for Equalized Coverage in GNNs"
_ACM.org/TheWebConf/2025/Conference — WWW 2025 Poster_

### Official Review · Reviewer_WBBd · 2024-11-10

**Novelty:** 5
**Technical Quality:** 5

**Review:**

This work focuses on fairness in uncertainty estimation for GNNs by introducing the concept of equalized coverage across demographic groups. They provide a theoretical framework for conditional coverage, enabling fair conformal predictions on graph data. The authors provide a theoretical analysis showing that fair GNNs can achieve consistent uncertainty bounds across groups. They validate these findings through experiments on several models on datasets.

Pros:

1, The problem presented is meaningful.

2, The authors provide a sound theoretical analysis.

3, The experiments appear sufficient for verification.


Cons:

1, The definition of "sensitive" vs. "non-sensitive" groups is unclear. Wouldn’t it be based on specific sensitive values (e.g., sensitive value = A vs. sensitive value = B)?

2, The work does not seem to use widely adopted fairness GNN datasets from recent years (e.g., NBA, Pokec-n, Credit, Bail, German) [1][2], which are generally encouraged for use.

3, The introduction to concepts and background is somewhat lengthy, making it less concise and intuitive.


Overall, I think this is an analytical study focused on a meaningful issue.

[1] Dai, Enyan, and Suhang Wang. "Learning fair graph neural networks with limited and private sensitive attribute information." IEEE Transactions on Knowledge and Data Engineering 35.7 (2022): 7103-7117.

[2] Dong, Yushun, et al. "Edits: Modeling and mitigating data bias for graph neural networks." Proceedings of the ACM web conference 2022. 2022.

**Questions:**

No additional questions.

**Reviewer Confidence:**

3: The reviewer is confident but not certain that the evaluation is correct

**Scope:**

4: The work is relevant to the Web and to the track, and is of broad interest to the community

---

### Official Review · Reviewer_zYQM · 2024-12-01

**Novelty:** 6
**Technical Quality:** 5

**Review:**

Conformal prediction on GNNs focuses on achieving coverage guarantees across the whole population, meaning uncertainty estimates might not accurately represent the coverage of individual sub-groups (e.g., race, gender). The authors introduce a framework for conditional coverage over subgroups and introduce the idea of “equalized coverage” guarantees across different groups. They provide theoretical results on coverage guarantees across demographic groups and identify that the probability of a sample being assigned to a particular group is the most important factor in upper bounding group coverage. They also use various utility and fairness metrics to empirically compare the performance of various existing GNN models.

Strengths:
1. Considering coverage guarantees on a group-specific level is an impactful goal, and the authors establish a framework for measuring group-specific coverage.
2. The authors provide interesting theoretical results to motivate work, although I would have liked to see more informal/english explanations of what the theorems mean and why they are important.
3. The empirical results shed insight on the comparative strengths of various “fair GNN” approaches and adhere to the theoretical bounds that the authors derive.

Weaknesses: Overall, the main takeaways from the empirical results section were relatively unclear to me. Both the writing and figures felt like they lacked important context and explanation necessary to make a compelling story surrounding the results.
1. It seems that the main takeaway from 4.2 is that GNNs that address structural bias are observed as being the most fair and therefore we should focus most on addressing structural bias if we want our GNNs to be fair. If the authors are going to make such a claim, it seems that it would be good to include more justification that these two fairness metrics are the correct ones to be using.
2. Section 4.2 also included many comparisons (e.g., “Graphsage demonstrates superior performance in terms of fairness compared to its counterparts”, “While [BeMap] may not consistently deliver top performance in all settings, its performance remains stable and reliable”) where it felt like there was not enough context to understand what the significance of the observation was or whether it should be surprising.
3. Figure 2 and Table 2 were both difficult to make sense of, and a disconnect between the figures and text further made it difficult to gain much insight from them. For instance, the only concrete text on Figure 2 is that all of the lines fall between the upper and lower bounds and that the figure “helps to identify trends and variations in coverage”, without explaining what these trends are or what they mean. If these are really the only takeaways, then the large number of lines and shapes does not seem necessary (especially since the shapes are quite difficult to visually distinguish, making it even harder for a reader to glance at the plot and interpret it on their own without guidance).

**Questions:**

1. Would connecting this work to existing literature on fairness criteria like parity make sense? The authors explain that this concept has not been looked at in GNNs, but there is extensive work on this type of criteria in other settings and it seems it would be valuable to connect these ideas more.
2. It seems like for each group, we want to jointly optimize for low coverage and inefficiency difference; why not have one metric for both to quantify how “badly” a model is performing for each group overall? Currently, one model could do better in coverage and worse in inefficiency, and then it is difficult to compare the two models or how they could/should be improved. More generally, more justification for why the currently-used metrics make sense would be helpful, if that is the main contribution of the paper.

**Reviewer Confidence:**

3: The reviewer is confident but not certain that the evaluation is correct

**Scope:**

3: The work is somewhat relevant to the Web and to the track, and is of narrow interest to a sub-community

---

### Official Review · Reviewer_TR3m · 2024-12-02

**Novelty:** 5
**Technical Quality:** 6

**Review:**

This paper investigates methods to provide unbiased uncertainty estimates for graph neural networks (GNNs) using conformal prediction. The authors present an interesting theoretical result that appears to contribute to the field.

Quality: The quality of the paper seems to me, to be very high. There is a serious theoretical contribution that, although I struggled with the details, was interesting and a useful result. The experimental evaluation seemed to be sound from my semi-lay viewpoint.

Clarity: The background section is well-written and provides a good overview. As someone who is superficially familiar with GNNs and quite familiar with conformal prediction and fairness, I found the background helpful. However, the concept of conditioning on a sensitive attribute is not clearly explained for a general reader.

The technical details presented in the paper are reasonably complex and challenging to follow, at least for me. Despite this, they might be acceptable for readers with a strong background in the subject.

I struggled to understand the connection between the experimental results discussed in Q1 and the theoretical framework outlined in Section 3.

Additionally, the paper does not explicitly define the term "equalized coverage."  While the introduction might imply its meaning, I believe a clear definition would enhance the reader's understanding.

Minor:
•	Assumption 1: “The conformity score S of the node v_j … remains unchanged *compared to the original permutation*”.

Originality: As a semi-lay reviewer who is familiar with conformal prediction and fairness, but only superficially familiar with GNNs and fairness, I cannot comment on significance compared to other work, choice of GNNs in experiment, choice of metrics, etc.

Significance:  The application of conformal prediction to GNNs is a useful and worthwhile topic to study. contribution. Extending this to fair conformal prediction is even more significant, as it addresses the crucial issue of fairness in machine learning models.

**Questions:**

What is the relationship between the theory in Section 3 and the results in Section 4.2?

**Reviewer Confidence:**

2: The reviewer is willing to defend the evaluation, but it is likely that the reviewer did not understand parts of the paper

**Scope:**

3: The work is somewhat relevant to the Web and to the track, and is of narrow interest to a sub-community

---

### Official Review · Reviewer_8ChT · 2024-12-02

**Novelty:** 6
**Technical Quality:** 6

**Review:**

#### **Overall Assessment**
The paper introduces a novel estimator of conformal prediction scores conditioned on equalized coverage across groups of nodes in a graph, defined by a sensitive attribute. The problem formulation, theoretical analysis, and experimental validation meet the high standards of *The Web Conference*. The work is significant for advancing fairness and uncertainty quantification in GNNs. The following detailed review highlights the paper’s strengths and areas for improvement.

---

### **1. Quality**
- **Strengths**:
    - The problem is well-formulated and addresses an important and timely challenge in fairness-aware machine learning.
    - The theoretical analysis is rigorous and provides strong support for the proposed approach.
    - Experimental results are comprehensive, validating the effectiveness of the proposed methods on multiple datasets and GNN architectures.
    - Notation, inputs, and outputs are clearly stated, facilitating readability and understanding.
- **Areas for Improvement**:
    - Computational cost and scalability are not adequately discussed. The only reference appears in line 251 (left column) when referring to related literature. In the revised version, explicitly state the computational complexity of estimating the conformal scores.
    - Claims in Section 4.2 about Figure 4 refer to distribution comparisons. Incorporating statistical tests to support these comparisons would enhance the rigor of the analysis.

---

### **2. Clarity**
The paper is generally clear and well-structured, but the following enhancements could improve clarity:
- **Typographical and Terminological Consistency**:
    - Correct the typo in line 263 (left column): "coMformity" should be corrected to "conformity."
    - Define \( \pi \) explicitly and ensure consistent usage (e.g., as permutation or statistics). In the appendix, remind readers what \( \pi \), \( A \), and \( X \) represent, particularly since \( A \) is commonly interpreted as an adjacency matrix.
- **Table Improvements**:
    - In Table 2, consider using colors or bolding to highlight the best, second-best, and third-best methods for clarity, as the table is large and dense.
    - For Table 1, the column “sensitive” contains a repeated entry, and the table layout exceeds the page margin. Remove this column and include its value in the caption, with a clear explanation of its meaning.
- **Evaluation Metric Definition**:
    - In Section 4.1, provide a precise definition of \( \text{Ineff}(a=x) \), as it is critical for understanding the evaluation.
- **Captions**:
    - Improve captions to make them more self-contained, enabling readers to interpret figures and tables without needing to cross-reference extensively with the text.

---

### **3. Originality**
The paper introduces a novel concept of equalized coverage and develops a corresponding estimator supported by strong theoretical analysis. The application of this concept to well-known GNN models in a node classification task is innovative and adds significant value to the field.

---

### **4. Significance**
The paper is highly relevant to the conference track and addresses a key challenge in fairness-aware machine learning for GNNs. The problem formulation and proposed solutions have potential applications in real-world scenarios, particularly in socially impactful domains such as hiring systems, healthcare, and financial services.

---

### **5. Reproducibility**
- The shared repository is currently expired, which hinders reproducibility. Ensure the repository includes:
    - Detailed instructions for running experiments.
    - Example configurations and datasets for replication.

---

**Questions:**

Please address the points outlined in the review.

**Reviewer Confidence:**

2: The reviewer is willing to defend the evaluation, but it is likely that the reviewer did not understand parts of the paper

**Scope:**

4: The work is relevant to the Web and to the track, and is of broad interest to the community

---

### Official Review · Reviewer_g15o · 2024-12-03

**Novelty:** 4
**Technical Quality:** 3

**Review:**

This paper focuses on investigating how the fairness-aware GNNs will benefit the equality in uncertainty estimation. Specifically, the authors conduct theoretical analysis to demonstrate that Fair GNNs will encourage equalized coverage. In addition, experiments are conducted on 5 datasets and 7 fair GNNs to verify it.

The reviewer believes this conclusion is very useful and appreciate the extensive experiments. Though conclusion is helpful, it is could be relatively easily inferred with the previous theories. And the unique challenges or insights in achieving equalized coverage are not covered. Hence, the reviewer encourage the authors to do more following works for publication.

Pros:
1. The work is well motivated. The problem of investigating equalized coverage is novel and required to be investigated.
2. The conclusion that Fair GNNs can be helpful for equalized odds is interesting.
3. Extensive experiments are conducted to verify the conclusion.

Cons:
1. The conclusion that  Fair GNNs can be helpful for equalized odds can be easily inferred based on the definition of fairness. Specifically, according to statistical parity, we will have $P(Y|S=0)=P(Y|S=0)$. The existing work generally use the uncertainty scores as $P(Y|S=0)$ for fairness constraints. Hence, it naturally benefit the equalized coverages.
2. It is highly suggested to investigate further investigate whether there are unique challenges for equalized coverage with fair GNNs.
3. The sensitive attributes in the experiments are not very realistic. It is suggested to conduct experiments on more realistic sensitive attributes.

**Questions:**

Please refer to the cons.

**Reviewer Confidence:**

3: The reviewer is confident but not certain that the evaluation is correct

**Scope:**

3: The work is somewhat relevant to the Web and to the track, and is of narrow interest to a sub-community